# Insights on Spark Plasma Sintering of Magnesium Composites: A Review

**DOI:** 10.3390/nano12132178

**Published:** 2022-06-24

**Authors:** M. Somasundaram, Narendra Kumar Uttamchand, A. Raja Annamalai, Chun-Ping Jen

**Affiliations:** 1Department of Manufacturing Engineering, School of Mechanical Engineering, Vellore Institute of Technology, Vellore 632014, Tamilnadu, India; somasundaram.m@vit.ac.in (M.S.); u.narendrakumar@vit.ac.in (N.K.U.); 2Centre for Innovative Manufacturing Research, Vellore Institute of Technology, Vellore 632014, Tamilnadu, India; 3School of Dentistry, College of Dental Medicine, Kaohsiung Medical University, Kaohsiung 80708, Taiwan; 4Department of Mechanical Engineering and Advanced Institute of Manufacturing for High-Tech Innovations, National Chung Cheng University, Chia-Yi 62102, Taiwan

**Keywords:** spark plasma sintering (SPS), magnesium composites, mechanical properties, corrosion, biocompatibility

## Abstract

This review paper gives an insight into the microstructural, mechanical, biological, and corrosion resistance of spark plasma sintered magnesium (Mg) composites. Mg has a mechanical property similar to natural human bones as well as biodegradable and biocompatible properties. Furthermore, Mg is considered a potential material for structural and biomedical applications. However, its high affinity toward oxygen leads to oxidation of the material. Various researchers optimize the material composition, processing techniques, and surface modifications to overcome this issue. In this review, effort has been made to explore the role of process techniques, especially applying a typical powder metallurgy process and the sintering technique called spark plasma sintering (SPS) in the processing of Mg composites. The effect of reinforcement material on Mg composites is illustrated well. The reinforcement’s homogeneity, size, and shape affect the mechanical properties of Mg composites. The evidence shows that Mg composites exhibit better corrosion resistance, as the reinforcement act as a cathode in a Mg matrix. However, in most cases, a localized corrosion phenomenon is observed. The Mg composite’s high corrosion rate has adversely affected cell viability and promotes cytotoxicity. The reinforcement of bioactive material to the Mg matrix is a potential method to enhance the corrosion resistance and biocompatibility of the materials. However, the impact of SPS process parameters on the final quality of the Mg composite needs to be explored.

## 1. Introduction

Over decades, metallic implants have been used extensively in the human body for different functionalities. However, there are some common shortcomings, such as the stress shielding effect, the need for secondary surgery to remove the implant, and the release of toxic ions on prolonging implantation. It gives a scope of research in biomaterials to enhance functionality by reducing the medical difficulty and increasing the aesthetical comfort of the host [1]. Biodegradable materials are identified to overcome the inabilities of non-biodegradable materials [2]. “Some of the biodegradable metals like Magnesium (Mg), Calcium (Ca), Zirconium (Zr), Zinc (Zn), Strontium (Sr), Silicon (Si), Manganese (Mn), Yttrium (Y) provide excellent biocompatibility, nutrients, biodegradation and required mechanical properties for the human body during [the] healing process” [3]. The bone healing process takes place in three steps (i) inflammation, (ii) repair, and (iii) remodeling. The mechanical integrity of the biodegradable implant must be maintained throughout the healing processes [3,4,5]; Mg is the potential lightweight material and is generally preferred for many structural and engineering applications by enhancing the mechanical properties [6,7]. 

Among the biodegradable materials, magnesium has more significant potential, since it has mechanical properties similar to the natural bone, which helps avoid the stress shielding effect. Moreover, it does not release any harmful content and is biocompatible. The great difficulty in using Mg-related implants is because of their higher corrosion rate and decrease in mechanical integrity throughout healing [4,8]. The difficulty can be overcome by optimizing material composition, selection, and processing and surface modification techniques. These are the factors that directly affect the biodegradable rate mechanical and biological properties, which give scope for the investigators to explore it [9,10]. Surface modifications are aimed to improve the life of the implants. It may be the coating of biocompatible or bioactive material and surface treatments [11,12]. Among the various techniques to enhance the function of the implants, the optimization or selection of material composition is believed to be a promising method. It paves a path for researchers to work on Mg-related alloys and composites [13,14]. Pure Mg is a highly reactive material in powder form at elevated temperature, low melting temperature, and brittle. These properties make the handling of Mg difficult. Care needs to be taken to maintain the working temperature.

Moreover, Mg exhibits a higher corrosion rate and less mechanical integrity in the host body throughout implantation [8,15]. Therefore, the focus is shifted to manufacturing Mg matrix composites rather than processing Mg and its alloys. Alloy is the mixture of two or more elements in which one element should be metal, but in a composite, metal may not be required to be present. Alloys are homogenous or heterogenous, but composites are only heterogenous. Due to the presence of metal elements, alloys are lustrous, which is not valid in the case of composites. The manufacturing flexibility of composites helps to attain desired mechanical properties by reinforcing different materials. Powder metallurgy is a potential technique for producing composite bioimplants [16]. However, the Mg powders are difficult to sinter in conventional sintering processes, as they form an oxide layer on the surfaces [17,18]; thus, maintaining proper inert gas atmospheric pressure requires sophisticated equipment.

The simultaneous application of pressure and electric field leads to a novel powder metallurgy technique called spark plasma sintering (SPS). The operating temperature of SPS is shallow compared to other conventional powder metallurgy (PM) processes. It helps to fabricate dissimilar materials with a homogenous structure without melting. The TiAl–Ti_3_AlC_2_ composite prepared through the SPS process shows a high relative density, but the addition of excessive reinforcement leads to agglomeration of the particle [19,20]. SPS of TiB_2_ and the ZrB_2_-SiCw-C_f_ system also shows relatively high density while adding nitrates [21,22]. The SPS of the Ti/TiB_2_-based composite allows the formation of in situ phases corresponding to significantly improving mechanical properties [23,24,25,26,27]. These studies show that the SPS process is used for high melting temperature materials, so processing temperature and oxide layer formation have been significantly reduced. The SPS is suitable for Mg-related alloys and composites, since it is a low-temperature process with a greater heating rate, undesirable reactions, and limited grain growth [28,29]. In recent times, SPS is also used for the processing of Mg-based bulk metallic glasses [30]. To the best of their knowledge, many research communities discussed the basic concepts, mechanisms, working, importance, and flexibility of SPS. However, the SPS of Mg and Mg-related composites is not reviewed extensively. The review aims to explore the application of SPS on Mg-related composites, especially the Mg composite with bioactive hydroxyapatite and ß tri-calcium phosphate. It also gives an insight into how the processing technique and reinforcement material alter the mechanical, biological properties, and corrosion behavior.

## 2. Spark Plasma Sintering of Mg Composite

Powder metallurgy is a manufacturing technique in which the final part can be obtained from a sequence of operations. In general, the sequence follows (i) the production of powder, (ii) compaction of powder, (iii) sintering of powder at an elevated temperature below the melting point, (iv) which is followed by post/secondary processing based on the requirement. Many researchers have been working on the fabrication of Mg composites to improve their functionality and properties. However, the optimal materials and process parameters must be maintained to attain a good response [31,32,33]. Teo et al. aimed to reduce the secondary phase formation of Mg-Zn-Ca alloy and Mg-Zn-Ca/SiO_2_ nanocomposite using a sinter less powder metallurgy technique followed by hot extrusion. The homogenous mixture is prepared using a ball milling process with a ball to powder ratio of 20:1. The powder blending is carried out at 200 RPM for 2 h without balls. The compaction is performed at 97 bar, and the sample is soaked for 1 h at 200 °C.

The hot extrusion is carried out at 200 °C, and the temperature is considerably lower than the eutectic transformation temperature [34,35,36,37,38]. The result shows good density, thermal and mechanical properties. It is concluded that temperature plays a vital role in controlling and refining the microstructure. It also controls the formation of secondary phases formed by X-Ray Diffraction (XRD) and Scanning Electron Microscopy (SEM) [39]. Therefore, there exists the need for low-temperature sintering. The study also shows the effect of the geometry of the die on power consumption. It reveals that an increase in the height of the graphite dies enhances the uniformity of temperature distribution and increases the power consumption rate [40].

SPS is a novel manufacturing technique for developing materials with different compositions by using a direct electrical current to rapidly consolidate powders in a brief period, using relatively high sintering pressure [41]. The high-speed consolidation happened because of the following phenomenon: plasma generation [42]; Joule heating [43,44]; pulsed current [45]; and mechanical pressure [46]. Figure 1a shows the schematic representation of spark plasma sintering. The subsequent list contains various components of the SPS process. Each has its importance and functionality: furnace cum pressing chamber; power supply for furnace; vacuum pumping system; pressing system; water circulation system; gas inlet system; control and instrumentation; temperature measurement tools; and mold sets [28]. Figure 1b shows the difference between SPS and conventional sintering. Figure 1c illustrates the flow of D.C. through the particles. There are four stages in the SPS process (i) maintaining atmospheric condition (vacuum/inert gas); (ii) compaction with the application of pressure; (iii) sintering by resistance heating; (iv) cooling [47]. The finite element simulation has been performed to understand the sintering mechanism, and the result shows that the heat conduction and Joule heating inside the sample are responsible sintering mechanisms [40]. The SPS of materials is completed based on various mechanisms, but the processes for the discharge and generation of plasma are extremely debatable within the research community.

In the scientific database, there is work that proves the presence of spark and plasma [48]. Some work also proved the “absence of arcing, sparking, and plasma in the whole sintering cycle” [49,50,51]. Compaction involves applying the pressure of the powders to gain density by eliminating air in between the particles. In SPS, the compaction is performed through a uniaxial hydraulic press. Typically, it should have a capacity of 250 kN, and it should also withstand the maximum sintering temperature of 2200 °C. The die is made up of either graphite or tungsten carbide. The instantaneous pressure is measured using a pressure gauge, and the temperature is measured/controlled with the help of a thermocouple and I.R. pyrometer. “Sintering refers to the process of firing and consolidating powders at temperatures lower than their melting point, where diffusional mass transport leads to bonding between particles and the formation of a dense body”. The entire process is carried out in the furnace cum pressing chamber. The chamber should have the capacity to maintain a vacuum or inert gas atmosphere [28]. The entire process is controlled by controllable parameters such as temperature, heating rate, atmospheric condition, holding time, and pressure. Table 1 shows details about the conventional compaction and sintering process. Table 2 shows the process parameters and corresponding mechanical properties of the SPS process. Table 1 and Table 2 clearly show that the process parameters such as temperature and sintering time are utilized far less often in the case of the SPS process. 

## 3. Microstructural Analysis

### 3.1. Characterization of Powders

A particle is a basic powder unit with various shapes and may consist of grains and intermetallic phases. The characterization of the particle can be achieved by measuring various parameters with different techniques. The quantitative data of powder are as follows: “particle size and its distribution, particle shape and its variation with particle size, surface area, interparticle friction, flow and packing, the internal particle structure, and chemical gradients, surface films and admixed materials [61]”. Among the various techniques, the microscopy technique is mainly used by researchers. Figure 2a reveals the broad particle size distribution and surface roughness of WE43 Mg alloy powder. In addition, the agglomeration of smaller particles is noted, which is termed as satellites. The backscattered electron mode (BSE) reveals the powder’s hexagonal dendritic structures, which is shown in Figure 2b [62]. The study reported that an Inductively Coupled Plasma Optical Emission Spectrometer (ICP-OES) is the instrument used to measure the chemical composition of the Cu-Cr-Mg powder [63]. The ICP-OES is also useful to measure or quantify the impurities in Mg such as Fe, Cu, Ni, and Co, which has a negative impact on the corrosion resistance on Mg-based materials. 

The characteristics of Mg-based alloy powder can be measured using an optical microscope, SEM, and XRD techniques. The XRD pattern helps to reveal phases of the powder [64]. XRD is also used to reveal crystallographic details of the sample. The effect of substituting aluminum (Al) and copper (Cu) on ƞ phase can be demonstrated. Furthermore, the nuclear magnetic resonance (NMR) helps identify the changes in the powder’s lattice parameters [65]. Figure 3a–c show the reinforcement particles of Al-xCu and its corresponding EDS analysis. It helps to have an insight into the calculation of the weight composition of the powder. 

### 3.2. Characterization of SPS Processed Sample

The microstructure of any specimen can be analyzed using various instruments which are worked based on various techniques. SEM analysis reveals the microstructure of the sample and helps to investigate the surface details of the tested specimen. The porosity and Mg enrichment region is identified through BSE, as contrasts of the image are based on an atomic number of elements in the specimen [50]. 

Energy-Dispersive Spectroscopy (EDS) helps identify the quantification of element distribution of the alloy or composite system. As for Mg in Ti-Mg alloy, initially, the strength of the alloy increases due to solid solution strengthening mechanisms, as the Mg dispersed in the Ti restricts the dislocation motion. A further increase in Mg content leads to a decrease in ductility, yield, and tensile strength due to the enrichment of Mg on Ti grain boundaries [50]. Zn–Mg/Mg–Zn–HAP (HAP—hydroxyapatite)-laminated composites are analyzed by SEM-EDS, showing no debonding in interfaces. It illustrates the possibility of manufacturing layered composites through SPS [57]. Field-emission SEM reveals the microstructure of Mg-TiB_2_ composite with the absence of porosity, micropores, and voids [67]. It also observed that only a tiny variation between the raw powder grain size and sintered sample indicates minimal or no grain growth. The microhardness of the composite decreases due to grain growth at high temperatures [68,69,70]. It can be reduced by the simultaneous application of pressure [67]. 

XRD results revealed the existence of magnesium oxide (MgO) and Mg because of SPS. Although the process occurs in vacuum conditions, oxidation happens because of the air between particles in the die, as shown in Figure 4 (where Mg10HA106 represents the Mg composite of Mg powder size of 106 µm reinforced with 10% HAP) [51]. XRD results reveal that an increase in the percentage of HAP leads to weak bonding between Mg-HAP, increasing the porosity. As the melting temperature of HAP is much higher than the sintering temperature, HAP is not sintered completely at that temperature (500 °C). Figure 5 shows the optical micrographs of the SPS sample of Mg-xHAP. Figure 6a–d show the optical micrographs of ZK61/xβ-TCP composites where the voids are present in sintered ZK61 alloy, which is not in ZK61/5β-TCP composite, as reinforcement fills the voids as shown in Figure 6b, but the agglomeration of β-TCP is observed in Figure 6c,d, while the particle boundary and the grain boundaries are observed in the optical micrographs. The recrystallization of Mg happens during the SPS process, as the grain size of Mg is much smaller than the raw powder observed in optical microscope images [51]. The volatilization of Mg content happened during sintering, which is identified as actual Mg content being less than the nominal value by XRD [61]. The precipitation of Mg is noted in SPS-processed Mg-related alloys. Although it is a rapid sintering process, precipitation cannot be eliminated [71].

In Mg–4Y–3Nd alloy, the formation of a stable β_e_Mg_41_Nd_5_ phase from the intermediate β_1_Mg_3_RE phase in the gas-atomized powder along with Mg is identified. The Electron Backscatter Diffraction (EBSD) image shown in Figure 7 reveals that the holding time has less or negligible influence on grain structure. However, the sintering temperature is the function of grain structure and residual strain. The uniform and thermodynamically stable grain structure are attained at 500 °C [71]. Fourier-transformed infrared (FTIR) spectroscopy facilitates the identification of the presents of chemical bonds. Similar spectra with different peak intensities are detected while investigating a Mg composite with rounded and cylindrical HAP reinforcement. It reveals the presence of HAP by bands of HPO^4−^ and PO_4_^3−^ groups [72,73]. The mechanical alloying process can significantly achieve atomic-scale alloying of Mg-related alloy because of its fracturing and cold welding [74]. 

## 4. Mechanical Properties

Hydroxyapatite (HAP) is one of the bioactive ceramics, which is highly recommended to use along with Mg to enhance its mechanical properties and bioactivity. The study shows that hardness increases with an increase in HAP percentage. The hardness of the Mg depends on the percentage of reinforcement [75,76]. The Young’s modulus, hardness, toughness, and ultimate compressive strength of the composites increase up to 15% of HAP reinforced in Mg-3Zn. Beyond 15% HAP, it shows a decreasing trend as there is clustering of Mg/HAP and HAP/HAP formed at the interface, creating porosity [58,76,77]. In conventional sintering, upon the addition of 5% HAP in Mg-3Zn, the mechanical properties show a decreasing trend [75]. The addition of excessive HAP provokes the agglomeration of particles, leading to unusual behavior of the Mg composite [60]. The compression strength of the Mg composite is improved as HAP is dispersed uniformly throughout the structure as the load is shared by uniformly distributed HAP reinforcement [78]. The mechanical properties of the Mg-5.5Zn/HAP composite have been tested based on ASTM E9-89a [79,80]. The yield compressive and flexural strength is increased by 43% and 21.8%, respectively, while adding 10% of nano-HAP on Mg-5.5Zn alloy [60]. The difference between the coefficient of thermal expansion and the shear modulus of Mg and HAP leads to residual stress development at the interface. It gives the path to necessary geometric dislocation, which is responsible for the increase in strength of the composite [75,81]. β TCP is also an excellent bioactive and biocompatible ceramic with good osteoconductivity [82]. The Mg/β TCP composite shows good load transfer behavior. The reaction between Mg and β TCP produces MgO, which increases the hardness of the composite. The oxide layer (MgO) is highly stable during SPS, since the process is carried out under ultra-high vacuum conditions [83]. Moreover, the Mg/β TCP composite forms the protective layer of calcium and phosphorus, which diminishes the degradation rate. The hardness and compressive strength of the ZK61/x β TCP increase with an increase in β TCP, to a maximum of 94.81 HV and 402 ± 9 MPa, respectively. The increase in β TCP tends to form local agglomeration, which increases the strength and hardness of the composite [59]. The slow and homogenous corrosion rate accomplished by SPS is due to the refinement of grain structure as the sintering takes place in a controlled environment with a lower temperature than the conventional sintering processes [62].

Furthermore, the relative density is the critical factor determining the SPS process’s effectiveness over conventional sintering processes. The conventional sintering process of Mg-xNb and Mg-xTa composites shows an average porosity of 2.5% (relative density of 97.5%) [31]. On the other hand, the relative density of the SPS-processed Mg composite was reported as 99.7%. It is possible to attain a relative density of greater than 99% in the SPS process, which is not valid in the case of conventional sintering [84]. The attainment of high density at lower processing time is possible in SPS, as the heating rate is high, which is correlated from the details given in Table 1 and Table 2. Apart from that material composition, process parameters also have a more significant influence on the mechanical properties of the composite, which need to be explored extensively to understand process-specific advantages rather than composition-specific properties. 

## 5. Corrosion Study

In general, the micro-galvanic coupling between cathodic and anodic areas leads to corrosion in Mg alloys in aqueous environments [85,86,87]. The ASTMG31–72 is followed to carry out an immersion test in simulated body fluid (SBF) and sample dimension of dia 15 mm of thickness 8 mm. The minimum corrosion rate and weight loss are exhibited by ZK61/5β-TCP as it forms a protective layer over the sample of 0.3165 mg/cm^2^/h and 28.87%, respectively [59,88]. The major problem associated with Mg is oxidation while processing. The Mg with a smaller particle size has a higher reaction rate than powder with a larger particle size. It leads to the formation of the MgO phase on the surface and may enhance the corrosion rate due to the formation of the oxide layer [84]. The SPS processing technique helps obtain better corrosion resistance of the material by enhancing the uniformity of the microstructure. The formation of MgO can be eliminated or significantly reduced by the inherent characteristics of the SPS process, such as faster densification at low temperatures [75]. The corrosion behavior of the Mg-5.5Zn/HAP composite has been analyzed with the help of electrochemical and immersion tests in simulated body fluid (SBF). The result shows good corrosion resistance in the Mg-5.5Zn/10HAP composite. The optimal amount of HAP needs to be dispersed in Mg to improve the density and corrosion resistance [60]. The grain growth is reduced to a greater extent with the added advantage of densification. The refinement of grains helps improve the ductility of Mg-related materials by overwhelming twin formation [89]. “Densification is due to particle rearrangement, localized deformation, bulk deformation, and neck growth”. The shrinkage rate of Mg alloy powder conforms to the effect of heating rate and initial particle size on consolidation [45,90,91,92,93]. It is observed that sintered Mg, Mg/10 β TCP, and Mg/20 β TCP show negative corrosion response, but the average corrosion current density is lower than extruded Mg [94].

Figure 8 shows the corroded ZK61 alloy and ZK61/xβ-TCP composites. According to the law of conversion, the corrosion rate is also analyzed by the amount of hydrogen gas released from the sample in SBF [95]. The evolution of hydrogen from the sample also reveals the corrosion rate in SBF and other corrosion mediums. A potentiodynamic study has been conducted to understand the corrosion behavior of the Mg-Zn/rHA or cHA composite. The Tafel plots help to derive the corrosion current and corrosion potential. The results show that apart from the reinforcement material, the shape of the reinforcement significantly contributes to enhancing corrosion resistance. The cylindrical HAP shows better corrosion resistance of 27% higher than rounded HAP reinforcement in Mg-3Zn [73]. The corrosion rate of rare elements (RE) containing Mg can be enhanced by cathodic activity and the formation of a protective layer over the surface [95,96]. Compared to conventional sintering, the lower sintering temperature is utilized for the SPS process, which benefits corrosion resistance [97]. Techniques such as implant retrieval histology analysis are not effective in examining the corrosion behavior of biodegradable implants on a real-time basis [98]. Although well-established techniques such as immersion test, evaluation of hydrogen gas, and electrochemical test are available to analyze corrosion behavior, there is a challenge in correlating results of in vitro and in vivo conditions [98]. The in vitro tests help examine the behavior of metal in simulated body fluid, but it does not provide details of the actual corrosion rate [99]. The real-time monitoring of the corrosion behavior of the biodegradable metallic implants is completed using the following models (i) Monitoring Local Changes Surrounding [100,101,102,103]; (ii) Fabricating an Intelligent Implant [104]; and (iii) Off-Clinic Point-of-Care Implant Monitoring [105,106]. Extensive research is required to monitor real-time corrosion and characteristics [107]. The corrosion testing conditions and corresponding corrosion rate are illustrated in Table 3.

## 6. Biocompatibility Study

The biodegradation behavior of implants tends to release metal ions in the surroundings, which would affect cell viability. The MTT assay of ZK61/β TCP is helping to perform cytotoxicity tests in L929 cells, and optical image of the tested sample is shown in Figure 9. The relative growth rate (RGR) is greater than 75%, indicating that the composite has no toxicity per ISO10993-5:1999 standard [110]. Moreover, ZK61/5β TCP shows good biocompatibility compared to other ZK61/xβ TCP composites, as corrosion has the tendency to release ions, which leads to a reduction in cellular activity and promotes cytotoxicity. 

Moreover, cell morphologies such as diamond, flattened spindle, or polygon represent good cells’ spreading [59]. The cytotoxicity of the Mg-5.5zn/10HA composite is also evaluated using L929 fibroblast cells. The RGR is higher than 80% as per the ISO10993-5:1999 standard [81]. Therefore, it is defined as non-toxic. The cell proliferation of the composite is increased due to the presence of the HAP content [74,111]. It also enhances the bone healing rate due to the release of ions such as Ca^2+^ and HPO_4_^2−^ from HAP [74,111]. The SPS technique is manufactured the porous Mg–Zn–Mn–Si–HAP composites. The bioactivity and biocompatibility of the composite are examined using MG-63 osteoblasts cell lines by cell culture, differentiation, and MTT assays. The result shows excellent biomechanical performance, and it also observed that the HAP is the response for enhancing cell adhesion and growth [112]. Many studies show better HAP and Si response results for enhancing their Mg-related composite/alloy [113,114]. The measure of optical density and DNA content helps obtain the outcomes of MTT assay and cell proliferation, respectively [115]. “The cell distribution growth on the sample surface was analyzed using a fluorescent microscope” [116]. Irrespective of the sintering condition, the coating of HAP enhances the corrosion resistance and cell viability [117,118,119].

## 7. Conclusions

The review article illustrates the various stages of the SPS process and its importance. It is evidenced that SPS is the practical process to attain full densification at the lower temperature range in a minimum time. It follows densification mechanisms such as particle rearrangement, localized deformation, bulk deformation, and neck growth. Compared to other powder metallurgy techniques, SPS helps produce good Mg composite products by the simultaneous application of pressure and temperature. However, it is also possible to achieve a higher heating rate, possibly reducing grain growth.

Mg has more affinity toward oxygen, so the processing of Mg in atmospheric conditions or at high temperature directs to the formation of oxides of Mg. So, the researchers are establishing the SPS of Mg composites under vacuum or inert gas atmosphere, which reduces the oxidation of Mg to a greater extent. 

The mechanical properties can be enhanced by reinforcing bioactive ceramics. In addition, the SPS process parameters play an essential role in enhancing grain structure by restricting grain growth, which enriches the composite’s mechanical properties.

The evolution of hydrogen gas can evaluate the corrosion rate of biodegradable Mg/Mg-related composite. For biomedical application, the amount of hydrogen should be within the following limit: “Ostwald solubility coefficient for hydrogen of whole blood in the human body is 0.018 (mL gas per mL medium)”. Therefore, this processing technique and percentage of reinforcement play a vital role in modifying the corrosion rate. Furthermore, it is required to correlate the in vitro study with the actual corrosion behavior of the implant. This challenge gives scope for future research.

The studies show that the Mg composites have good biocompatibility behavior with enhanced bioactivity. The excellence of Mg composites is greatly influenced by their material composition (weight proposition, size, shape of fiber and reinforcement), microstructure, and surface properties. Moreover, corrosion trends release metal ions, affecting cell viability. Although the Mg composites show promising results, there is a long way to use Mg composite as a bioimplant. The stages are as follows: (i) in vitro test; (ii) in vivo test; (iii) product shape formation; (iv) animal models; (v) human volunteers; and (vi) final product. It paves the way to work on Mg/Mg alloys/Mg composites for biomedical applications.

Many studies show the effect of reinforcement on composites while processing through the SPS process. However, only a few studies show the effect of SPS process parameters on responses, which need to be explored.

## Figures and Tables

**Figure 1 nanomaterials-12-02178-f001:**
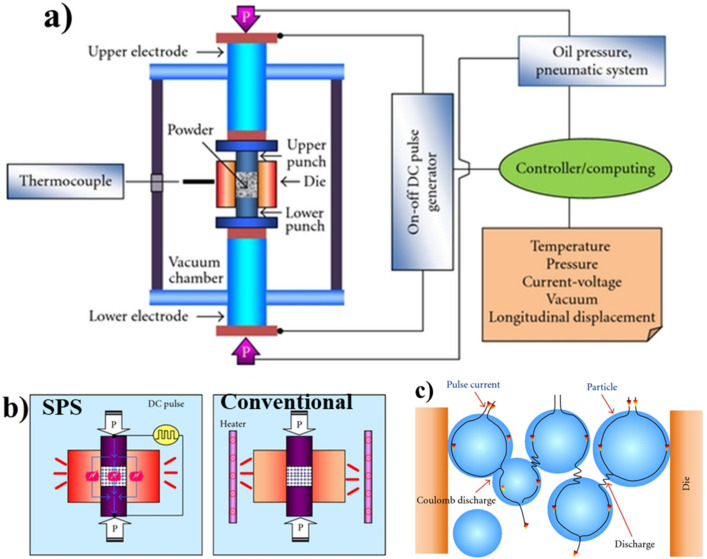
Schematic representation of (**a**) Spark plasma sintering; (**b**) Comparison between SPS and conventional sintering; (**c**) D.C. pulse current between particles (adapted) [47].

**Figure 2 nanomaterials-12-02178-f002:**
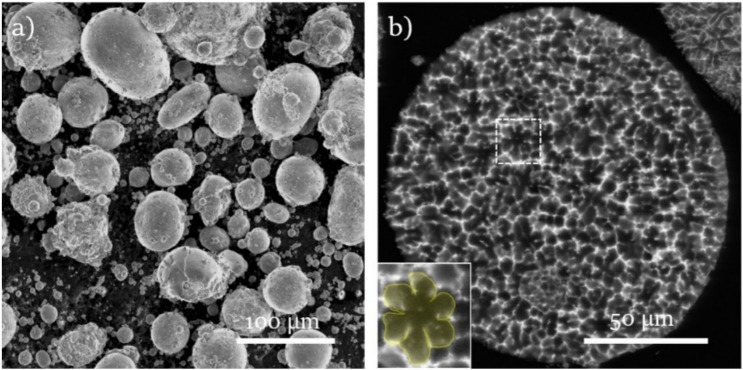
Gas atomized WE43 powder: (**a**) SEM micrograph; (**b**) SEM/BSE micrograph to show the cross-section [62].

**Figure 3 nanomaterials-12-02178-f003:**
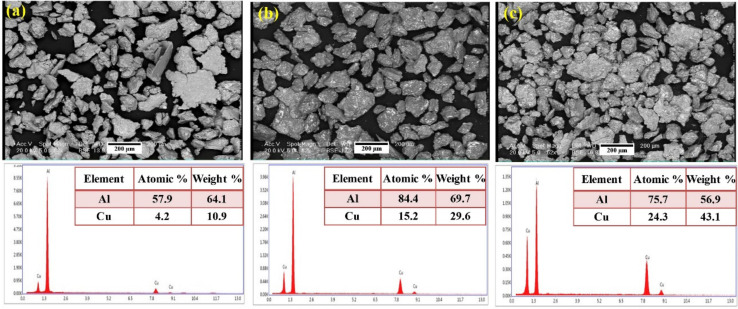
SEM images and EDS analysis of Al-xCu powders consisting of various amounts of Cu: (**a**) x = 20, (**b**) x = 33 and (**c**) x = 50 [66].

**Figure 4 nanomaterials-12-02178-f004:**
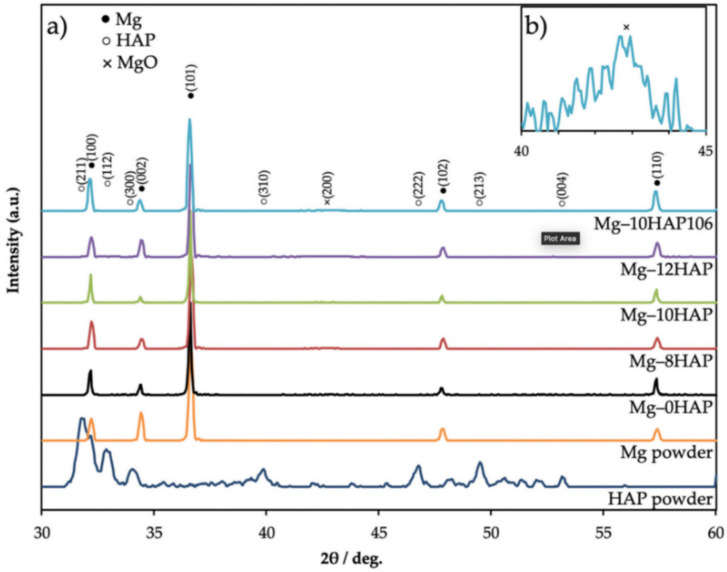
XRD results of Mg, HAP powder, and Mg-xHAP composites (**a**)XRD pattern; (**b**) Extended view of Mg-10HAP106 within 40–45° [51].

**Figure 5 nanomaterials-12-02178-f005:**
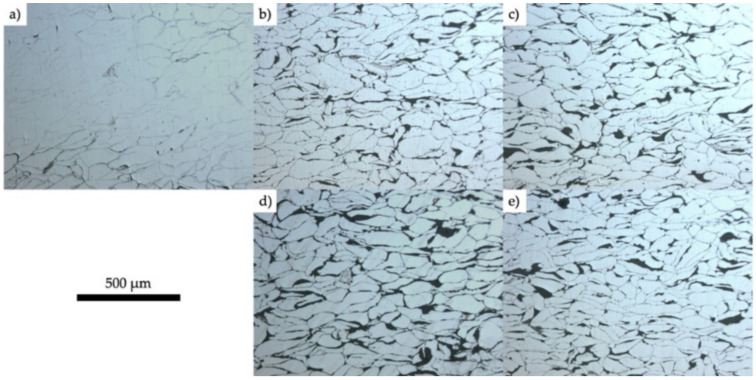
Optical micrographs of (**a**) Mg-0HAP, (**b**) Mg-8HAP, (**c**) Mg-10HAP, and (**d**) Mg-12HAP, and (**e**) Mg-10HAP106 after sintering using SPS system [51].

**Figure 6 nanomaterials-12-02178-f006:**
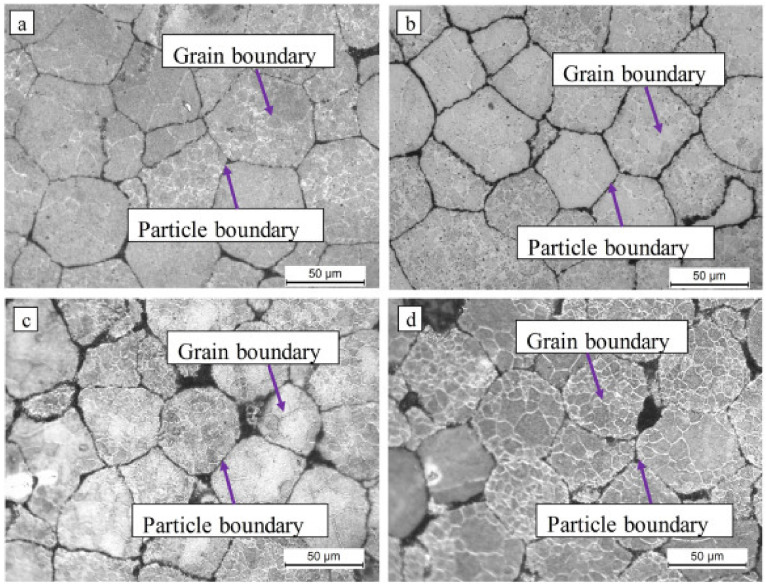
Optical micrographs showing the microstructures of samples: (**a**) ZK61, (**b**) ZK61/5β-TCP, (**c**) ZK61/10β-TCP and (**d**) ZK61/15β-TCP [59].

**Figure 7 nanomaterials-12-02178-f007:**
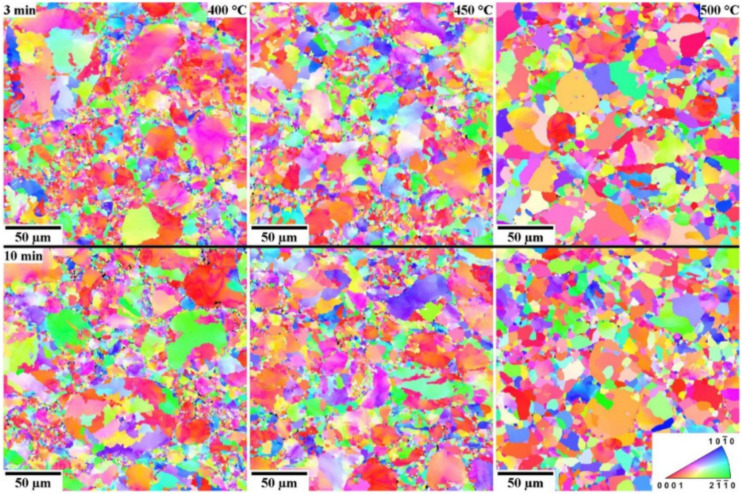
EBSD image of Mg-based composite at different sintering temperature and time [71].

**Figure 8 nanomaterials-12-02178-f008:**
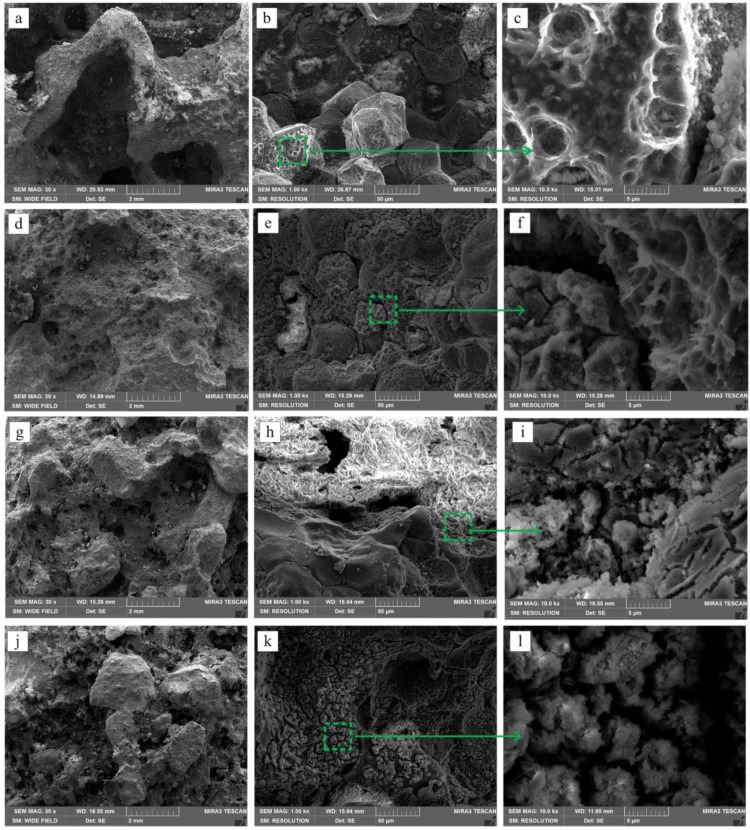
The surface morphologies of composites after 15 days in SBF solution after removal of corrosion products: (**a**–**c**) ZK61, (**d**–**f**) ZK61/5β-TCP, (**g**–**i**) ZK61/10β-TCP and (**j**–**l**) ZK61/15β-TCP [59].

**Figure 9 nanomaterials-12-02178-f009:**
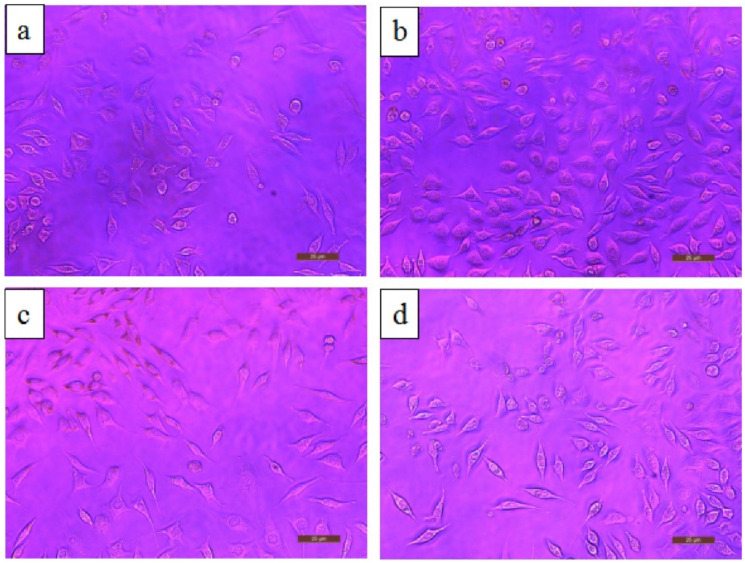
Optical micrograph of L-929 cells cultured in 100% extraction mediums for 72 h (**a**) ZK61, (**b**) ZK61/5β-TCP, (**c**) ZK61/10β-TCP and (**d**) ZK61/15β-TCP composite [59].

**Table 1 nanomaterials-12-02178-t001:** Process parameters of conventional compaction and sintering of Mg composite.

S. No.	Type of PM	Mesh	Mixing/Blending	Compaction	Sintering	Ref.
1	Mg-SiCp SiCp of 10, 20, 30%	Mg—0.23 mm, SiC—0.26 mm	Ball milling	100 Tons Hydraulic press, Pressure 400 MPa, 60 S	Solid-state sintering Argon atm. 460 °C 30 min	[6]
2	AZ91D (92%), Tugsten carbide (2, 4, 6%), Graphite (6, 4, 2%)	Up to 50 microns	Ball milling Time = 1 h Speed = 200 RPM	Pressure 200 N/mm^2^	500 °C at an increased heating rate of 5 °C/min for 1 h and cooled in the furnace	[7]
3	Mg-3Zn-1Ca and Nb	Zinc, Niobium, Mg—325 mesh Calcium—6 mesh	Ball milling Ball to powder = 20:1 Time = 1 h Speed = 200 RPM At inert argon atm	Hydraulic press Cylindrical die 100—500 (400) MPa, dwell time 8–10 min dia 10 mm and 15 mm height ISO 13314	Two-step sintering at 380 °C and 610 °C Heating rate 300 °C/h soaking time 6 h	[32]
4	Magnesium and Silicon carbide (0, 4, 8, 12%)	Mg—50 to 290 microns SiC—30 to 50 microns	High energy ball milling	Hydraulic press attached in the universal testing machine	Muffle furnace	[52]
5	Mg (80 wt %), Zn (19 wt %), and Ca (1 wt %) powders SiO_2_ (1 wt %) nanoparticles		Ball milling Stainless steel ball Ball to Powder = 20:1, Time = 7 h; Speed = 200 RPM Argon atm	50 tons, 35 mm dia and 40 mm length	No sintering Hot extrusion at 200 °C for 1 h Extrusion ratio 20.25:1 The final part of 8 mm	[39]
6	Pure Mg and Mg with TiO_2_ (1.5, 2.5, 5%)	60–300 microns	In planetary ball milling machine without the ball; Time = 1 h; Speed = 200 RPM	Pressure 960 N/mm^2^ dia 32 mm and length 40 mm	Hybrid Microwave sintering (1.1 kW, 2.45 GHz by V.B. Ceramic consultants) 1. 500 °C for 4 h 2. 400 °C for 1 h	[33]
7	AZ91D and B_4_C (5, 10,15, 20%)	AZ91D—10 and B_4_C—60 microns	High-energy ball milling Balls of diameter = 10 mm; Ball to Powder = 20:1; argon atm; 2 wt % of stearic acid with methanol to avoid oxidation	-	-	[53]
8	Mg (99% pure) with irregular shape Nanohydroxyapatite (0, 2, 4, 8%)	80 microns	Hydroxyapatite nanoparticles were extracted from the bovine bone	The cylindrical mold of 8 mm dia The pressure of 350 MPa and maintain it for 2 min	250 °C for 2 h and then to 550 °C for another 2 h in a vacuum furnace (1 × 10^−4^ Torr)	[54]
9	Mg and naphthalene particle	Mg—85 to 100 microns Naphthelene—300—350 microns	-	Uniaxially pressed at 125 MPa pressure to 8.3 mm diameter cylindrical samples	Heated in a hot air oven at 120 °C for 24 h to sublime naphthalene. Afterwards, the samples were sintered at 550 °C for 2 h under argon atmosphere	[55]
10	Mg, Ta, Nb	Mg—60 to 220 microns Ta—5 to 120 Nb—2 to 15	Planetary ball mill PM 400-Retsch; Ball to Powder = 20:1; Time = 9 h; Speed = 200 rpm argon atm	Uniaxially pressed at a pressure of 760 MPa into cylindrical compact	sintered at 610 °C for 3 h	[31]
11	Mg, Fe, Zn, Ca granular, Nano-hydroxyapatite, carbamide particles—(30, 40, 50%)	-	-	Uniaxially pressed by using a hydraulic press at a pressure of 90 MPa for 2 min; diameter 36 mm and thickness 6 mm	Tubular furnace under controlled high-purity argon atmosphere. 250 °C with a rate of 3.5 °C/min and holding at this temperature for 5 h to evaporate carbamide particles, and then at 500 °C with a rate of 4 °C/Min and a dwell time of 2 h at this final temperature	[56]

**Table 2 nanomaterials-12-02178-t002:** Mechanical properties of spark plasma-sintered Mg composite.

Type of PM	Mesh	Mixing/Blending of Powder	Die/Sample Size	Compaction Pressure (MPa)	Sintering Temperature, Time (°C and min)	Composition	CYS (MPa)	UCS (MPa)	Failure Strain (%)	Hardness	Yield Strength (MPa)	Tensile Strength (MPa)	E (GPa)	Fracture Toughness MPa.m ^1/2^	Ref.
ZK61 alloy/rGO nanoplatelets	75 microns		diameter 20 mm thickness 5 mm	60	520, 6	0.5 rGO/ZK61 (X/Y)	143	368	11.7	69.5	-	-	-	-	[48]
						0.5 rGO/ZK61 (Z)	147	438	19.5	70.2	-	-	-	-	
AZ91 alloy/Ti powder	Mg < 74 microns Ti < 37 microns	Ball milling; 50 RPM for 2 h in argon atm; 10:1	diameter 25 mm thickness 12 mm	75	500, 20	AZ91-5Ti	185	442.2	25.3	-	-	-	-	-	[49]
						AZ91-10Ti	206.2	437	24.4	-	-	-	-	-	
						AZ91-15Ti	255.6	439	21.8	-					
Mg powder (purity 99.50%); Nano-HAP (purity 95%)	Mg = 180 microns	Planetary ball milling; 500 RPM for 10 min; argon atm		50	500, 10	Mg—10HA	207.2	-	-	-	-	-	-	-	[51]
						Mg—12HA	137.8	-	-	-	-	-	-	-	
Mg powder 99.88%; Zn powder 99.99%; HAP powder 98.5%	Mg =75–150 microns; Zn = 23 microns; HA < 60 microns	GN-2 High energy ball milling; agate balls of 2:1 ratio; 400 RPM	Die dimensions Height = 40 mm; inner dia = 30 mm; outer dia = 50 mm	40	390, 5 min	Zn–10Mg	161	-	-	-	-	-	4.78	48.3	[57]
						Mg–5.5Zn–5HAp	115	-	-	-	-	-	4.66	31.9	
						Two-layered laminated composites	222	-	-	-	-	-	7.11	55.2	
						Three-layered laminated composites	235.3	-	-	-	-	-	7.86	69.7	
Mg powder (99%); Zn powder (99.8%); HAP nanopowder of cylindrical shape	Mg = 130 ± 15 microns; Zn = 30 ± 5 microns; HA = height 180 ± 20 nm; dia 83 ± 8 nm	Wet precipitation method	Grade 2333 graphite	80	Two-stage sintering; 450, 5 and 500, 5	Mg-3Zn/5HA	-	230.1 ± 5.9	14.6	-	111.8 ± 3.2	-	-	126.8 J/m^3^	[58]
						Mg-3Zn/10HA	-	242 ± 2.7	16.1	-	120.1 ± 3.9	-	-	165.8	
						Mg-3Zn/15HA	-	257.3 ± 5.4	17.5	-	131.7 ± 2.6	-	-	143.7	
						Mg-3Zn/20HA	-	207.2 ± 4.3	13.2	-	97.5 ± 1.8	-	-	89.2	
ZK61/x βTCP	ZK61—45–75 μm; βTCP < 38 μm	Ball milling for 10 h with 5 min pause time of every 30 min at 400 RPM with a ratio of 2:1	Dia 30 mm × 20 mm	40	500	ZK61/5βTCP	-	338 ± 13	19.5 ± 0.6	-	-	-	10.51	-	[59]
						ZK61/10βTCP	-	368 ± 5	18.3 ± 0.5	-	-	-	10.18	-	
						ZK61/15βTCP	-	402 ± 9	17.8 ± 0.3	-	-	-	10.67	-	
Mg-Zn/HAP	Mg <150 μm; Zn < 25 μm; HAP = 60 nm	Ball milling; 400 RPM at 3 h in argon atmosphere; ball to powder ratio of 4:1	Graphite dia of 20 mm	40	480, 5 min		-	-	-	-	-	-	-	-	[60]

Note: HA/HAP—Hydroxyapatite; G.O.—Graphene Oxide; Ti—Titanium; Zn—Zinc; βTCP—beta-tricalcium phosphate.

**Table 3 nanomaterials-12-02178-t003:** Corrosion behavior of Mg composites.

Material	Method of Testing	Sample Size	Atmospheric Condition (°C)	Medium	Duration (h)	Rate of Corrosion (mm/year)	Ref.
monolithic Zn–10 Mg and Mg–5.5Zn–5HAp composites	Immersion test (ASTMG31–72)	Ø9 × 7	37.0 ± 0.5	simulated body fluid (SBF)	72	0.69 (Zn–10 Mg) 8.23 (Mg–5.5 Zn–5 Hap	[57]
Mg-xHAp composite	Immersion test	4 × 4 × 8	37	Hank’s solution	120 (5 days)	HAp improves the rate of corrosion	[51]
ZK61/xβTCP	Immersion test (ASTMG31–72)	Ø15 × 8	37.0 ± 0.2		360 (15 days)	0.3165, 0.3365, 0.3415 mg/cm^2^/h	[59]
Mg-3Zn/rHA or cHA	Potentiodynamic study	1 cm^2^ (opening in surface)	37 ± 1	m-SBF	14 days	5.263 ± 0.26 (Mg-3Zn) 4.084 ± 0.24 (Mg-3Zn/15rHA) 3.225 ± 0.22 (Mg-3Zn/15cHA)	[73]
Mg–MgF_2_ and WE43-MgF_2_	Immersion test	1 cm^2^	37	SBF	14 days	Mg-MgF_2_—0.346 ± 0.047 WE43-MgF_2_—0.875 ± 0.062;	[108]
Mg–1Al–Cu/xGr	Immersion test (ASTMG31–72)	8 × 6 × 2 mm	37	SBF	10 h	Rate degradation is low in 0.18 Gr, but it increases with Gr	[109]

## Data Availability

The data will be shared upon request from the authors.

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
