# Peer review of "Insights on Spark Plasma Sintering of Magnesium Composites: A Review"

_nanomaterials, 2022, doi:10.3390/nano12132178_

Round 1
Reviewer 1 Report
For authors
The review article is based on a review of the literature, but I miss the authors' experience and their results in researching of magnesium alloys and Mg-composites.
There are few graphic representations, tables, and figures in the article, especially images of microstructures.
Abstract is deficient, does not explain the main topics in the article. It is not clear which magnesium alloys (by chemical composition) it describes and which Mg composites (what is the matrix and what is the hardening phase).
The abstract needs to be thoroughly redesigned.
The abbreviations need to be explained in the order as stated in the article so e.g. abbreviation ydroxyapatite (HAP) is explained in line 218, but it is mentioned for the first time in line 184.
Error in line 198: Error! 198 Reference source not found.5 shows the optical micrographs of SPS sample of Mg-xHA. 19.
A clear overview of Mg alloys and phases in these alloys, and hardening compounds is missing.
A clear overview of Mg alloys about mechanical properties, dansity, grain size and other characteristic with exact numbers and not just in the proportion of differences is missing.
ISO10993- 51999 standard in line 316 must be cited
Conclusion: statement in conclusion line 356 and 357: »The ability of Mg composites is based on their material composition, microstructure, and surface properties« I would like to have a little more explanation and discussions.
The literature review is extensive and includes more recent sources.
Overview of alloys and in more detail their mechanical properties is missing.
It would be good to add more graphic and photographic material.
Reviewer 2 Report
This article contains a review in the area of composites based on magnesium. This area finds considerable attention in the field biomedical applications, since besides mechanic properties in particular the degradation product of the main component, magnesium ions, are not toxic at all (by contrast, magnesium ions are even essential for the body in large quantities). The reader is introduced into the subject with concise and informative explanations, and accordingly this review is also suited for readers who are not experts in the field but would like to obtain an overview in this area, which also includes the spark plasma sintering technique.
The review is fluently written and provides an appropriate selection of articles, and experimental parameters as well as conclusions are provided. It is built up in a logic sequence of chapters, which include key aspects of the related materials, such as characterization, mechanical properties, corrosion behavior and biocompatibility.
All in all I do not have major problems with this manuscript, however, one error should be corrected: On p. 4, last paragraph, line 198 – 199 the message “Error! Reference source not found” arises. It seems that something is not working with the dedicated reference.
Also, in the first line of the Introduction (in total line 28), the statement “Metallic implants have been used extensively in the human body in recent days” is not appropriate since such implants have not been used extensively only in recent days but for decades. In fact, this is not evident from this review so this should be considered in the revised version.
Author Response
Please see the attachment. The responses and marked revision are included.

Reviewer 3 Report
Comments on the paper:
The manuscript titled: "Insights on Spark Plasma Sintering on Magnesium Composites: A Review" sounds interesting giving an insight into the microstructural, mechanical, biological and corrosion resistance of spark plasma sintered Magnesium composites. As written it is almost ready for publications.
Before publication there are few remarks to consider.
1) In line 198, Please correct the error, "Error! Reference source not found"
2) In Figure 5, please put the explanation for e) part of the Figure 5, in the figure title, and also put the "d" into the brackets "(d)"
3) In 4. Mechanical Properties, in line 226, please reformulate the statement: "But the HAP crosses the optimal rate; it provokes agglomeration of HAP"
4) In line 231, "The yield compressive and flexural strength is increased by 43% and 21.8%, respectively." Please, provide more information on that statement.
5) In line 246, Please provide more information on "comparatively less sintering temperature"
Author Response

(The authors gave the same response as above.)

Reviewer 4 Report
It is necessary to correct or correct the following places in the text:
1) The sentence located on lines 63-66 of page 2 of section 1 needs to be made more understandable and unambiguous.
2) The proposal located on lines 66-69 of page 2 of section 1 is also desirable to be corrected, since magnesium powders do not sinter in air, but sinter in vacuum or in an inert gas environment.
3) Correct editor's note on lines 198-199 of page 4.
4) In section «7. Conclusion» to remove the text located on lines 331-335 from the article, as it duplicates the text in other sections. And lines 336-341 are combined into one thematic paragraph (output).
5) From the section «7. Conclusion» remove references to literary sources (line 352) or move this discussion, for example, to section «6. Biocompatibility study».
Round 2
Reviewer 1 Report
The authors supplemented the article accordingly.